# Giant Tumefactive Perivascular Space: Advanced Fusion MR Imaging and Tractography Study—A Case Report and a Systematic Review

**DOI:** 10.3390/diagnostics13091602

**Published:** 2023-04-30

**Authors:** Renata Conforti, Raffaella Capasso, Donatella Franco, Carmela Russo, Fabio Oreste Rinaldi, Giovanna Pezzullo, Simone Coluccino, Maria Chiara Brunese, Corrado Caiazzo, Ferdinando Caranci, Fabio Tortora

**Affiliations:** 1Department of Precision Medicine, University of Campania “Luigi Vanvitelli”, 80138 Naples, Italy; renata.conforti@unicampania.it (R.C.); fabiorinaldi1.fr@gmail.com (F.O.R.); giovanna.pezzullo@unicampania.it (G.P.); simone.coluccino94@gmail.com (S.C.); ferdinando.caranci@unicampania.it (F.C.); 2Department of Radiology, CTO Hospital, Azienda Ospedaliera dei Colli, 80131 Naples, Italy; dott.ssacapasso@gmail.com; 3Unit of Neuroradiology, Department of Neurosciences, Santobono-Pausilipon Children’s Hospital, AORN, 80129 Naples, Italy; russocarmela84@gmail.com; 4Department of Medicine and Health Sciences “V. Tiberio”, University of Molise, 86100 Campobasso, Italy; mariachiarabrunese@gmail.com (M.C.B.); caiazzocorrado@gmail.com (C.C.); 5Unit of Neuroradiology, Department of Advanced Biomedical Sciences, Federico II University, 80138 Naples, Italy; fabio.tortora@unina.it

**Keywords:** perivascular spaces, magnetic resonance, giant/tumefactive perivascular spaces, advanced magnetic resonance

## Abstract

Perivascular spaces (PVSs) are small extensions of the subpial cerebrospinal space, pial-lined and interstitial fluid-filled. They surround small penetrating arteries, and veins, crossing the subarachnoid space to the brain tissue. Magnetic Resonance Imaging (MRI) shows a PVS as a round-shape or linear structure, isointense to the cerebrospinal fluid, and, if larger than 1.5 cm, they are known as giant/tumefactive PVSs (GTPVS) that may compress neighboring parenchymal/liquoral compartment. We report a rare asymptomatic case of GTPVS type 1 in a diabetic middle-aged patient, occasionally discovered. Our MRI study focuses on diffusion/tractography and fusion imaging: three-dimensional (3D) constructive interference in steady state (CISS) and time of fly (TOF) sequences. The advanced and fusion MR techniques help us to track brain fiber to assess brain tissue compression consequences and some PVS anatomic features as the perforating arteries inside them.

## 1. Introduction

Perivascular spaces (PVSs) are pial-lined spaces [1,2] surrounding small penetrating cerebral arteries as they go through the subarachnoid space to the brain tissue [1,3,4,5,6,7,8]. They form a network throughout the brain which is part of the glymphatic system [9], playing a role in the lymphatic drainage of metabolites and having an immunological function [9,10]. Enlarged PVSs are expansions of the normal millimetric ones but with a short axis greater than 2–5 mm [3,4,5]. The vast majority of PVSs are incidental, clinically unimportant, usually asymptomatic findings [8] and considered “leave me alone lesions” [1]. PVSs represent physiological findings [10] that are routinely seen in magnetic resonance imaging (MRI) in 100% of subjects [2]. PVSs are ubiquitous in elderly patients, being slightly dilated in healthy ones and enlarged in 32% of cases [5,6,7,8,9]. Such “normal” PVSs, usually microscopic in size, can be observed on MRI as small curvilinear or oval structures [2], with cerebrospinal fluid (CSF)-like signal intensity in all pulse sequences [2,5], no enhancement after paramagnetic contrast medium injection, and with the normal signal intensity of the surrounding brain parenchyma [1].

Rarely, PVSs may have larger dimensions (>1.5 cm) with mass effect [1,7]; in these cases, they are known as giant/tumefactive PVSs (GTPVSs) [10], setting an important diagnostic challenge [2]. GTPVSs are relatively uncommon; only one retrospective review by Salzman et al. shows 37 cases until 2004 [10]. GTPVSs are poorly understood entities whose pathogenesis is unclear. Furthermore, little is known about their clinical manifestations, MRI appearance, and management [1]. Both clinical manifestations and MRI appearance of GTPVSs may depend on their anatomic brain location, so they are divided into three main types [3,4,5,6]. Type 1 appears along the lenticulostriate arteries entering the basal ganglia through the anterior perforated substance; type 2 is found along the paths of the perforating medullary arteries, and type 3 appears in the mesencephalon-thalamic region [1,2,3,4,5,6,9]. Cortical PVSs (type 2) are lined by a single layer of pia, whereas two layers accompany the lenticulostriate (type 1) and collicular arteries (type 3) form [9]. Type 2 and type 3 GTPVSs are the most frequent [1], whereas type 1 GTPVSs are described only in 3% of cases. We report the case of a type 1 GTPVS for their relatively rare reports [1,2] and because, until now, a type 1 GTPVS had never been studied with fusion MRI and diffusion tensor imaging (DTI) in an asymptomatic patient.

We use MRI gradient-echo (GRE) 3D constructive interference in steady state (3D CISS) sequence, as suggested by previous authors [11], to investigate the fine morphological features of GTPVSs with high signals and extremely high spatial resolution. Then we fuse CISS and MR angiography time of fly (TOF) sequences to better show the subtle anatomy of perforating arteries within the GTPVS. Moreover, following other authors’ previous MRI studies [4,8,12,13,14], we provide the utility of DTI to define the abnormal distorted neuroanatomy of the region.

## 2. Case Report

A 48-year-old man presented with a one-year history of diabetes mellitus (DM) type 1, diabetic polyuria, and polydipsia. Two months earlier, he had a fever (40 °C) for many days, partially resolved with broad-spectrum antibiotic therapy. One week later, he developed acute urinary retention. The abdominal ultrasound study showed up portal vein thrombosis and prostatic hypertrophy with calcifications. The Computed Tomography (CT) study of the abdomen showed a fluid/superfluid collection in the paramesocolic groove, extending caudally, reaching the recto-vesical cavity, where it appears partially organized. From the spleen-mesenteric confluence to the hepatic hilum, the vena porta was occluded until its lobar and segmentary branches suggesting thrombophlebitis. The liver was swollen, with irregular borders and inhomogeneous, for portal perfusion defects. The spleen appeared to increase in size (craniocaudal length: 150 mm) with an accessory spleen (about 12 mm) above the superior pole. Multiple confluent and increased-in-size pathologic abdomen-pelvic and liver hilum lymph nodes were seen. The abdominal aorta was diffusely atheromatic. Blood analysis revealed infection: C-reactive protein (CRP) 20 mg/dL, white blood cells (WBC) 15,000/mm^3^, and the most are neutrophils (10,700 mm^3^), ESR 119 mm/hr, and blood sugar level 335 mg/dL. An adjunctive blood test confirmed a bacterial infection (streptococcus constellatum) responsible for pelvic abscess. The neurological examination was normal. The CT brain study showed a left capsule-lenticular-striatal homogenously hypodense rounded area with moderate mass effect, without peripheral contrast enhancement. Within the cystic-like area, a linear enhancing structure appeared after iodate contrast medium injection, suggesting a vessel (Figure 1a,b).

The MRI study was performed using a General Electric Signa Voyager 1.5 T scanner (GE AG, Muenchen, Germany) with a 32-channel head coil. Written informed consent was signed by the subject before the study. DTI was acquired using a single-shot spin-echo echo-planar imaging (SE/EPI) with Repetition Time (TR) of 11,500 ms, Time to Echo (TE) of 83.6 ms; matrix of 96 × 96; field of view (FOV) of 240 × 240; slice thickness 2.5 mm, *b*-value 1.000 s/mm^2^; number of excitations (NEX) 1; in 32 noncollinear directions; acquisition time 7:08′. 3D CISS sequence was acquired with TR 6.26 ms, TE 2.70 ms; matrix 384 × 384; FOV 180 mm; flip angle 40°; slice thickness 0.75 mm, slice per slab 144; slice oversampling 6.7. For a complete MRI study, other sequences were: 3D Cube fluid-attenuated inversion recovery (FLAIR) on sagittal plane (TR 5002 ms; TE 117 ms; Inversion Time TI 1487 ms; matrix 244 × 230; FOV 25.8 × 25.8; slice thickness 1.5 mm; NEX 1; acquisition time 6:13′); 3D susceptibility weighted angiography (SWAN) on axial plane (TR 77.6 ms; TE 43.9 ms; matrix 320 × 256; FOV 24.5 × 17.2; flip angle 15°; slice thickness 2.2 mm, NEX 1; acquisition time 5:29′); axial plane 3D fast spoiled gradient echo (FSPGR) T1 before and after gadolinium intravenous injection (TR 8.1 ms; TE 3.7 ms; TI 24 ms; matrix 280 × 280; FOV 24 × 24; flip angle 12; slice thickness 1.2 mm; NEX 1; acquisition time 3:43′); axial plane SE/EPI diffusion-weighted images (DWI) b1000 (TR 7822ms; TE 69.9 ms; matrix 128 × 128; FOV 24 × 24; slice thickness 4 mm; NEX 3; acquisition time 1:34′); axial plane Fast Spin Echo (FSE) 3D Cube T2 (TR 2002 ms; TE 140 ms; matrix 320 × 320; FOV 20 × 20; slice thickness 0.8 mm; NEX 2; acquisition time 4:29′) and axial plane 3D TOF (TR 22 ms; TE 6.8 ms; matrix 288 × 288; FOV 23 × 17.3; flip angle 20; slice thickness 1.4 mm; NEX 1; acquisition time 5:03′).

The MRI study confirmed the up-mentioned CT features, showing a cystic-like parenchymal area 3 × 2.8 cm in diameter and 3.0 cm^3^ volume, isointense to the CSF in all sequences, partially surrounded by a close thin area of gliosis, no contrast enhancement, but with better evidence of the linear structure within the area, compressing the neighboring brain structures (Figure 1). The 3D CISS sequence showed the “ribbon-like” structure inside the cystic-like lesion that the fusion image (CISS + TOF) confirmed to be a perforating artery with a redundant course (Figure 2). DWI and apparent diffusion coefficient (ADC) maps showed no signal restriction (not shown). Based on radiological studies, the diagnosis suggested a GTPVS.

DTI preprocessing data included brain extraction using the FMRIB Software Library (FSL), version 6.0.0, program BET [1] and correction of eddy current distortion and motion using the FSL program EDDY. DTI registration entailed combining an affine image transformation (from native diffusion space to native 3D T1 space) using the FSL program “flirt” with a non-linear warping (from native 3D T1 space to standard MNI space) using the FSL program “fnirt”. DTI post-processing was made with GE workstation software “Ready view”. The left and right thalami were separately used as seed regions for DTI based on the Harvard–Oxford Atlas. For each seed, the external capsule ipsilateral target masks were created from the JHU FSL atlas. Probabilistic tractography was performed to estimate the strength and the most likely location of a pathway between the respective seed and target areas. Voxel-wise estimates of the fiber orientation distribution were computed using the FSL program “BEDPOSTX” (Bayesian Estimation of Diffusion Parameters Obtained using Sampling Techniques). Then, probabilistic tractography was performed using the FSL program “Probtrackx2” with the “classification target” option enabled and all default settings: 5000 samples/voxel, step length 0.5 mm, curvature threshold 0.2. The “loop check” option was also enabled to avoid streamlines looping back on themselves. In this way, for each hemisphere, each voxel in the seed region was seeded with 5000 samples of streamlines that migrated according to local probability density functions. An exclusion mask for the contralateral hemisphere was also specified to restrict the generated tracts to the ipsilateral hemisphere with respect to the seed mask. This resulted in two tractographic maps in standard MNI space (one per each hemisphere region) where only streamlines that propagated to a specific target mask were counted. Because the number of estimated streamlines for each pair of seed and target region is also dependent on the size of the seed region, a connection probability index was calculated as the ratio between way total (streamlines that reached the target) and total streamlines (5000× seed) voxels [2]. Fractional anisotropy (FA), mean diffusivity (MD), and radial diffusivity (RD) were extracted from tractographic maps generated from tractography and from bundles of fibers (atlas mask in MNI space) surrounding PVSs: uncinate fasciculus (UF) L and R, external capsule (EC) L and R, anterior thalamic radiation (ATR) L and R, and cortico-spinal tract (CST) L and R.

ROI analysis from principal bundles surrounding the PVS revealed FA, MD and RD values summarized in the following table (Table 1). The right connection probability index, calculated as the ratio between way total (streamlines that reached the target) and total streamlines (5000× seed) voxels, resulted in a decrease on the left side (2.37) with respect to the right side (3.35), which implies a lower density due to the GTPVS mass effect. Quantitative analysis on the left side from the tractographic map revealed an increase of FA value = 0.3742, with respect to the right side (FA value = 0.359), and left decrease of both MD value = 0.866 × 10^−3^ mm^2^/s (MD value= 0.911 × 10^−3^ mm^2^/s) and RD value of 0.698 × 10^−3^ mm^2^/s, with respect to the right side (0.744 × 10^−3^ mm^2^/s). Moreover, the total number (N) of left-side streamlines also decreased (438,160) with respect to the right side (total N of streamlines = 630,193) (Table 2).

## 3. Discussion

Historically, PVSs were first described by the German pathologist Rudolf Virchow and French anatomist Charles Philippe Robin, so they were also called Virchow–Robin spaces [9].

The glial limiting membrane and the outside of the vessel wall form the PVS boundaries. These pial folds separate the surrounding brain tissue and CSF while creating an enclosed space filled with interstitial fluid (ISF); on standard MRI, PVS and CSF appear isointense in all sequences, but the quantitative MR imaging signal values are different [2]: this suggests that PVSs are filled with ISF rather than CSF [9], although there is a free fluid transfer between them [2,15].

PVSs form a complex intraparenchymal network distributed throughout the brain, connecting the cerebral convexities, basal cisterns, and ventricular system, but the PVS does not communicate directly with the subarachnoid space or CSF. Literally, a GTPVS is a giant dilatation of PVS, greater than 1.5 cm in size, and thought to be the result of blocking the outlet of ISF for any reason [9]. It is reported that tortuous arterial branches may cause focal distortion of the overlying cortex that may, in turn, obstruct small tracts traversed by perforating vessels that connect the perivascular and subarachnoid spaces [3]. Until the advent of MRI, PVSs have been reported as incidental findings at autopsies and classified as normal anatomical variants and/or a result of aging brains in the literature. With widespread MRI, they became more visible and considered a normal variant in the literature. However, after the recognition of the cerebral glymphatic system and related excretory roles, they started to be evaluated in more detail.

While not completely understood, the PVS network plays an important role both in providing drainage routes for cerebral metabolites and in maintaining normal intracranial pressure [9]. They are thought to drain ISF into the lymphatic vessels of the head and neck. PVSs expose indwelling macrophages to a variety of different antigens, which are ultimately phagocytosed and presented to B-cells and Helper T-cells, resulting in their activation [10]. PVSs are part of the cerebral glymphatic system, in which CSF-ISF exchange occurs within the brain parenchyma, probably mediated by aquaporin 4 water channels and a substantial amount of ISF and cerebral metabolites, such as amyloid beta, exits the brain via connections between the PVS and leptomeningeal vessels. In this way, intravenous paramagnetic contrast agents can also enter the PVS [9].

Different opinions have been expressed regarding the cause of PVS dilatation [2], but it remains incompletely understood [2,15]. The most reliable postulated hypotheses provide increased arterial wall permeability due to vasculitis, defects in the drainage of the brain’s ISF into the ventricles as a result of increased intraventricular CSF pressure [10,15], drainage defects of the brain’s ISF due to lymphatic obstruction [2,15], inflammatory activity leading to cellular infiltration and edema [15], and spiral elongation of penetrating blood vessels secondary to hypertension [5,6,10]. Current studies demonstrated that enlarged PVSs are associated with compromised blood–brain barrier integrity [16]. Other authors ascribe the dilatation of PVSs to chronic ischemic diseases [2].

Even if clinically asymptomatic, our patient was diabetic. Therefore, we looked for some meaningful link between a hyperglycemic condition and PVS dilatation. Some authors [17,18] report that GTPVSs in their DM patients could be a manifestation of small vessel disease (SVD) due to diabetes [2,18]; moreover, they go into the pathophysiological process of glymphatic fluxes impairment in the case of DM, discovering the consequent accumulation of molecular waste within the PVS and activation of inflammatory response and neurovascular disruption. Until now, little is known about how DM may affect the glymphatic system impairing it. Some authors report that DM induces both micro- and macro-vascular damage, which increases the risk of developing SVD resulting in dilatation of the PVS [17].

The clinical presentation of GTPVSs varies according to the extent of cystic-like lesion expansion in terms of both their space-occupying effect and location [15]. GTPVS are, as in our case, fortuitously discovered because they usually do not induce any clinical abnormality [10,12,19]. Some authors report that approximately 50% of patients with GTPVS suffer from headaches [15]. Others do not describe headaches or dizziness [2], even with GTPVS compressive effects, as in our case [1]: dislocation of the adjacent corticospinal tract, abnormal diffusion tensor metrics, and abnormal tractography [8]. The lack of systematic data on GTPVS may lead to diagnostic confusion and variable non-evidence-based management of patients who have these brain anomalies. Type 2 and 3 GTPVS are the most associated with symptoms, even if Kwee et al. [1] suggest that type 2 GTPVS uncommonly may cause clinical manifestations. However, when type 2 GTPVS are spacers and disseminated, they may cause symptoms of parkinsonism and/or dementia [1,4]. Pyramidal tract symptoms caused by type 2 GTPVS mass effect are rare, with only one reported case [1]. Notably, type 2 GTPVSs have also been observed in patients with neurofibromatosis [18,20] and psychomotor retardation, which may suggest a genetic cause in these patients [1].

Type 3 GTPVS seem to be more likely to cause clinical manifestations in at least 75.0% of patients because their location can cause noncommunicating hydrocephalus, which occurs slowly over a long period of time by compression of the cerebral aqueduct or third ventricle and/or symptoms due to local mass effect on the mesencephalon [1,7,9,10]. In the literature, only six patients are reported to have a GTPVS in the posterior fossa, a notable finding which may therefore be considered an atypical location; mostly, in these cases, it is mandatory to acquire post-gadolinium injection T1w images to exclude the presence of cystic neoplastic and non-neoplastic lesions [1,21].

An adequate differential diagnosis is needed between GTPVSs and other cystic pathologic processes [2,22]; consequently, it is fundamental that their MRI features deep knowledge. GTPVSs must be discriminated from intracranial cystic neoplasms and, moreover, from arachnoid cysts. The cyst content, the mass effect, and the cystic lesion location (especially for arachnoid cysts) are fundamental factors to distinguish these lesions from PSV. Chronic lacunar infarcts are hardly differentiated from PVSs and barely from their shape. Inflammatory pathologies, such as multiple sclerosis, are easily differentiated in their acute-enhancing phase. Finally, the differential diagnosis between parasitic cysts (cryptococcosis and neurocysticercosis) and GTPVSs can be made by evaluating cyst content and contrast enhancement [7]. GTPVSs can be seen on MR imaging as round, oval, tubular, or curvilinear well-defined and cyst-like structures with smooth margins, isointense to CSF [2,10], usually following vascular distribution. Even though they are filled with ISF, PVSs usually follow CSF signals on all MRI sequences, but quantitative studies reveal a statistically significant difference between a PVS and CSF, which is generally unrecognizable by the human eye. They neither show restricted diffusion on DWI, which helps to differentiate them from recent ischemic lesions, nor calcification, hemorrhage or high protein content, unlike other congenital, infectious or tumoral cerebral cystic lesions [9]. GTPVSs can present as a solitary lesion or in clusters [10] with a symmetric distribution. Tumefactive types are usually solitary, and they do not enhance with intravenous gadolinium-based contrast agents [9].

According to Lim et al., Rawal et al. and Cerase et al., perilesional T2/FLAIR hyperintensity is noted in 80% of patients [3,18], as shown in our case with the hyperintense feature in brain tissue around the GTPVS.

There is no clear consensus on the radiological feature of the GTPVS neighboring brain parenchyma modifications [1,2]. Otherwise, some authors report that in 75% of cases, MR images should not show any signal-intensity abnormality of the adjacent brain tissue; moreover, it is reported that 25–32.3% [1,9] of patients show hyperintense rim and 5–10% have a hyperintense area on neighboring brain tissue [9]. This feature appears to be more common in type 2 GTPVS (33.9% of patients) [1].

The cause of GTPVS surrounding white matter (WM) T2/FLAIR signal change is uncertain. Someone postulates accelerated WM ischemic changes resulting from GTPV compressive effects on adjacent parenchymal vessels or chronic mechanical stress on the brain arterioles from high blood pressure [1]. However, another theory suggests that the high signal may be gliotic modification [2,19].

In this study, in agreement with previous reports, we underline the essential role of the 3D CISS sequence to enable the estimation of brain water content and to evaluate fluid-filled spaces and their 3D observation, clearly revealing the fine morphological features of GTPVSs. Moreover, GTPVSs in the basal ganglia, as in our case, are described in the literature as irregularly sized, with some parts dilated and others markedly narrowed, and arteries seen within them are identified as perforating arteries. Sometimes GTPVSs communicate with CSF in the basal cistern crossing the fiber tracts of the internal capsule [11]. Type 1 GTPVSs have two histologic components: a vessel traversing the brain from the pia and a fluid-filled space surrounding the vessel [23]. The recently described characteristic “linear sign” was observed also in our study, in both 3D CISS and TOF images, indicating the presence of GTPV-traversing vessels [3] being thin straight or slightly curved lines cutting across cystic lesions. Particularly, in the case of a PVS ≥ 5 mm, a 3D CISS sequence is recommended to carefully identify thin linear low signal structures within the cystic lesions [3,23]. Using TOF, we confirm previous observations that lenticulostriate arteries are an important source of “linear sign”; moreover, it gives better specificity and accuracy than T2w sequences alone [23]. We use TOF/3D CISS fusion images to increase the MRI sensibility of recognizing the two different GTPVS histological components [11] (Figure 2).

We also performed a 3D reconstruction based on the FSPGR T1 sequence [24,25], showing the tight mutual connection between the GTPVS and penetrating vessels crossing it (Figure 1).

The stability in size over time and appearance of GTPVSs are critical for their diagnosis, although a few cases of progressive GTPVSs have been reported. However, the progression rate is not as fast as that of tumors [9]. If the imaging findings are pathognomonic for a GTPVS, no further workup is needed, and biopsy should be avoided, as there is a risk of hemorrhage by damaging the penetrating vessel in the GTPVS [1].

Although GTPVS cases have been previously reported in the literature, only one study reports DTI changes and tractography abnormalities, but not in the case of a GTPVS type 1 in a middle-aged asymptomatic patient [12].

We follow the MRI study pattern of previous authors [4,12,26], proving the utility of diffusion and tractography MRI in understanding the abnormal neuroanatomy of this condition [8,12]. Furthermore, in our case, we try to explain the correlation between MRI pathologic data that shows a very large GTPVS and a fiber pathway subverted with the absence of neurologic symptoms.

FA describes to what extent water diffusion occurs anisotropically, and it is thought to be related to WM integrity since FA reduction may indicate WM damage [12]. On the other hand, increased FA in the ROI, as we have founded on the left (especially ATR and UF), may suggest increased diffusion anisotropy and structural coherence that may occur from local pathologic conditions as increased in myelination, axonal diameter, axonal density, and/or increased directionality [8].

The increased FA within the tracts compressed by GTPVSs may be due to subtle mass effect (especially on the L ATR), leading to a reduction of extracellular fluid accumulation (and increased both diffusion anisotropy and structural coherence), as suggested by other authors [12]. The changes in DTI and the increase in FA are due to the reduction of unrestricted water in the extravascular extracellular spaces (EES) because of the mass effect of GTPV [8].

In agreement with previous studies [4,8,12], we also discovered reduced values of both MD and RD on the left side with respect to the right one, suggesting diminished magnitude that, one more time, reveals alterations in the extracellular volume of both grey matter (GM) and WM, as a consequence of decreased EES. The total number (N) of right streamlines = 630,193, whilst the N of left streamlines = 438,160. ROI analysis from principal bundles surrounding the GTPVS revealed FA, MD and RD values summarized in the following table (Table 1).

Moreover, also in our case, tractography showed compression, displacement and thinning of fiber bundle track, the inferior orbitofrontal (IFO) (Figure 3), as assessed by the difference between the total number of streamlines significantly reduced on the left.

Indeed, the DTI showed left displacement and decreased number of WM fibers from principal bundles in the region of interest surrounding the GTPVS with respect to the opposite side. Figure 4 illustrates the axial directionally encoded color FA map, as reported by other authors. The GTPVS mass effect is responsible for both thinning, compression and distortion of WM fibers and modifications of FA, with an apparent decrease in their number rather than real damage [12,19]. The absence of symptoms in our patient agrees with as described previously [8,12], confirming the earlier hypothesis so that tensor abnormalities do not correlate with any clinical signs or symptoms [8] and do not necessarily predict any pathological condition [12].

## 4. Conclusions

GTPVSs are rare and can be misdiagnosed with central nervous system tumors; however, their distinctive features on imaging, especially using TOF fused with 3D CISS sequences, are useful to demonstrate central perforating artery inside them, leading to a more accurate diagnosis, avoiding inappropriate surgical procedures.

We describe, for the first time, a clinically asymptomatic patient with unilateral type 1 GTPVS, despite compression of the neighboring parenchyma, abnormal diffusion tensor metrics, and abnormal tractography. We confirm that also type 1 GTPVs may induce diffusion tensor and tractography changes due to brain mass effect and compression. The absence of symptoms may be due to their slow growth, which displaces rather than damages brain tissue without deterioration of brain function.

## Figures and Tables

**Figure 1 diagnostics-13-01602-f001:**
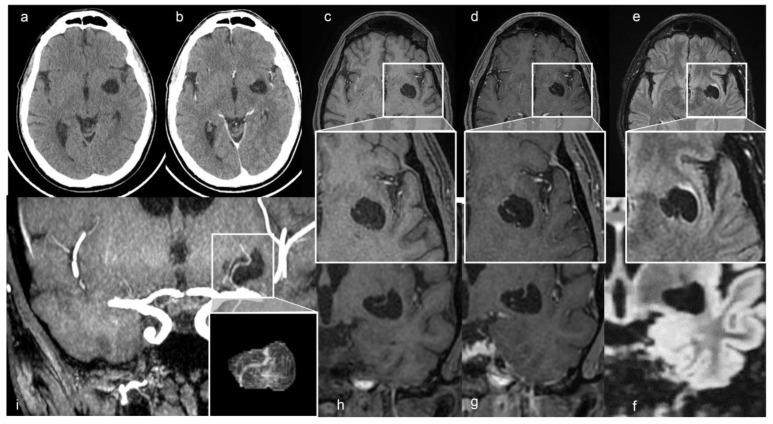
(**a**) Axial non-contrast and (**b**) enhanced CT scan shows a left cystic-like intraparenchymal area, hisodense to CSF, suggestive of a GTPVS; in (**b**) note linear enhancing feature inside the cyst. Axial T1, T1 with gadolinium and FLAIR MRI sequences, (**c**–**e**) and magnified views; coronal T1, T1 with gadolinium and FLAIR MRI sequences (**f**–**h**) of GTPVS. Note in (**e**) the thin T2 hyperintensity surrounding the GTPVS, responsible for mild compression of neighboring parenchyma. (**i**) Coronal maximum intensity projection (MiP) TOF arterial 3D surfaces reconstruction of the GTPVS.

**Figure 2 diagnostics-13-01602-f002:**
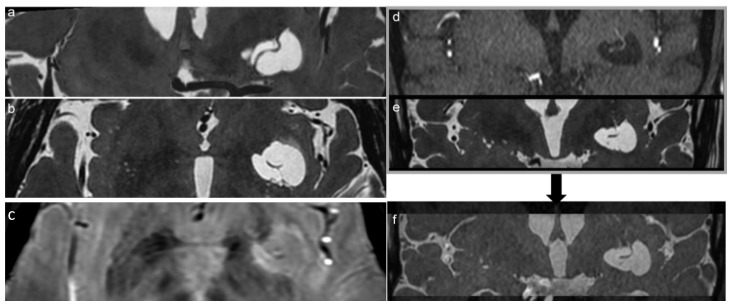
Coronal and axial reconstruction of the 3D CISS sequence shows the irregular shape of the GTPVS with multiple linear vascular-like images inside (**a**,**b**). (**c**) 3D SWAN on the axial plane better demonstrates a linear vascular structure within the GTPVS. (**f**) notes the clearer image of the vessel inside the GTPVS resulting from the fusion of d-TOF and e-CISS images. (**d**) Coronal TOF and (**e**) coronal CISS images are fused together in figure (**f**): notes the clearer image of the vessel inside the GTPVS resulting in the fused image.

**Figure 3 diagnostics-13-01602-f003:**
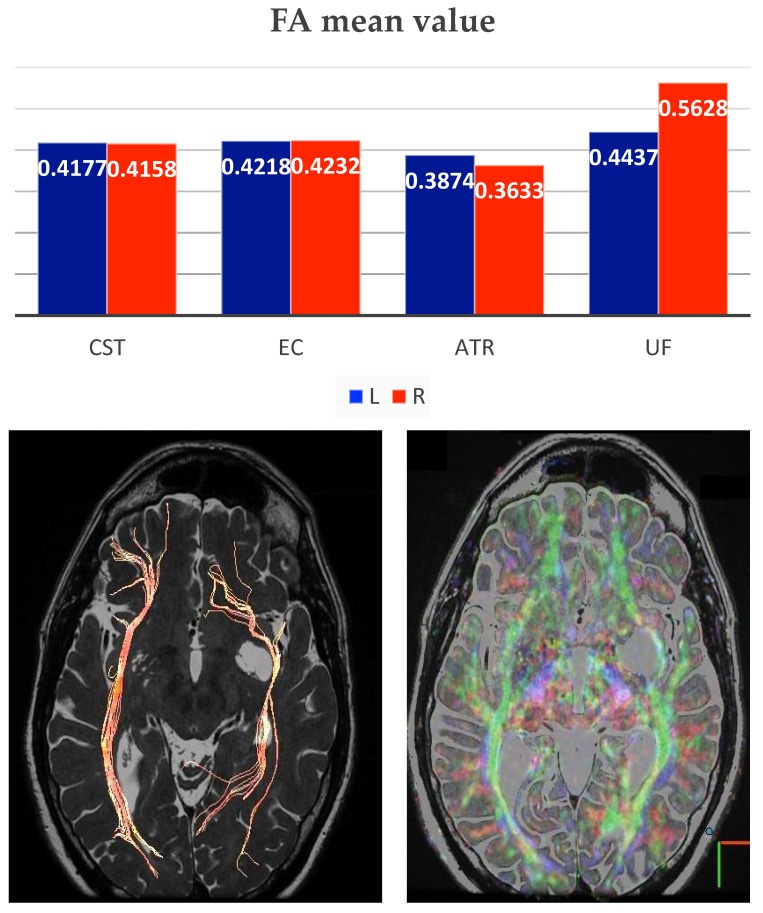
Histogram shows the Fractional Anisotropy (FA) mean value of ROIs analysis in specific structures (Cortico-spinal tract: CST; external capsule: EC; anterior thalamic radiation: ATR; uncinate fasciculus: UF), right side value in red and left value in blue. In the images below, it is possible to see DTI VR Fiber Tracking and DTI 32 directions CISS fusion showing displacement and thinning of the fiber bundle, in this case the inferior frontal-orbital (IFO).

**Figure 4 diagnostics-13-01602-f004:**
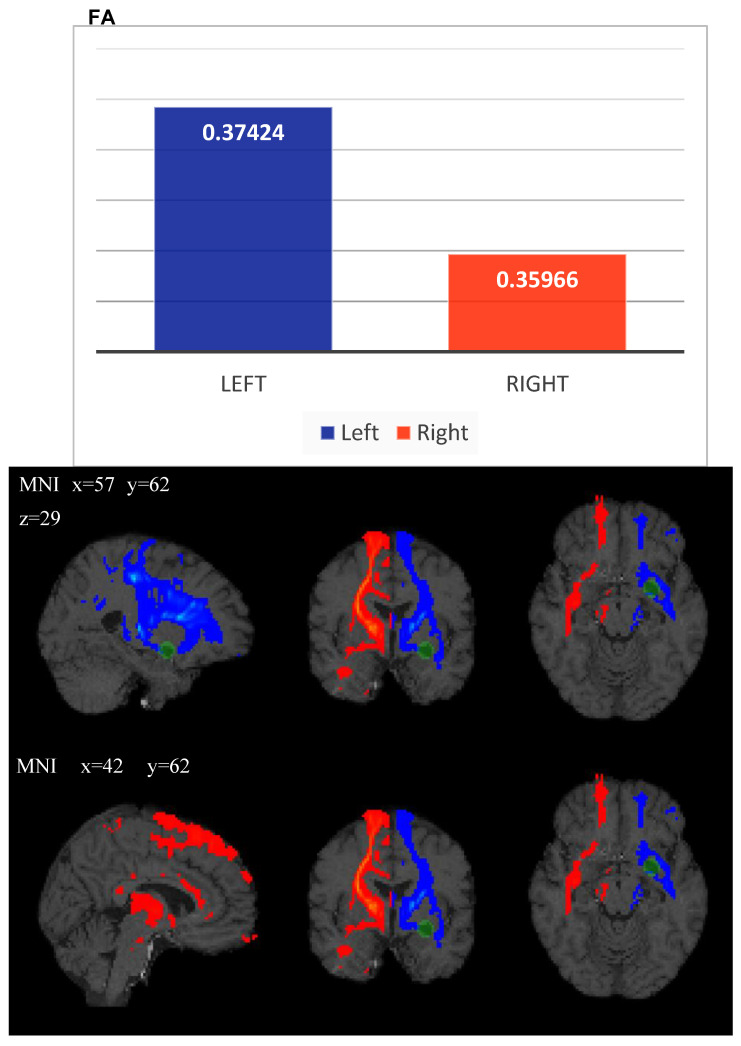
Histogram of FA mean value (left blue and right red) of tractography maps obtained between Thalamus-like seed mask and external capsule-like termination mask. FA: Fractional Anisotropy; L: Left; R: Right. Probabilistic tractography from Thalamus as seed region and external capsule as target mask for both sides (blue on the left side and red on the right side). Green ROI highlights the GTPVS. MNI: Montreal Neurologic Institute coordinate system.

**Table 1 diagnostics-13-01602-t001:** FA, MD, RD of ROIs analysis right and left side: mean value and standard deviation in brackets of cortico-spinal tract: CST; external capsule: EC; Anterior thalamic radiation: ATR; uncinate fasciculus: UF. FA: Fractional Anisotropy; MD: Mean Diffusivity; RD: Radial Diffusivity; ROI: Region of Interest.

	N Voxels		Mean (sd) FA		Mean (sd) MD		Mean (sd) RD	
	L	R	L	R	L	R	L	R
CST	10,548	9814	0.4177 (0.198689) (0.198689)	0.4158 (0.195867)	0.00081 (0.000334)	0.00084 (0.000379)	0.00063 (0.000360)	0.00065 (0.000405)
EC	692	729	0.4218 (0.123302)	0.4232 (0.127692)	0.00080 (0.000159)	0.000811 (0.000098)	0.00062 (0.000178)	0.00061 (0.000128)
ATR	12,118	9987	0.38748 (0.175312)	0.3633 (0.173499)	0.000935 (0.000478)	0.00102 (0.000571)	0.00075 (0.000492)	0.00084 (0.000585)
UF	49	47	0.4437 (0.190764)	0.5628 (0.117045)	0.00110 (0.000704)	0.00074 (0.000090)	0.00088 (0.000731)	0.0004 (0.000112)

**Table 2 diagnostics-13-01602-t002:** Mean value (left and right) and standard deviation in brackets of tractography maps obtained between Thalamus-like seed mask and external capsule-like termination mask. FA: Fractional Anisotropy; MD: Mean Diffusivity; RD: Radial Diffusivity; L: Left; R: Right.

	N Voxels		Mean (sd) FA		Mean (sd) MD		Mean (sd) RD	
	L	R	L	R	L	R	L	R
Tractography map	12,596	12,650	0.37424 (0.184479)	0.35966 (0.187262)	0.000866 (0.000315)	0.000911 (0.000358)	0.000698 (0.000351)	0.000744 (0.000395)

## Data Availability

Data is contained within the article.

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
