# Peer review of "Giant Tumefactive Perivascular Space: Advanced Fusion MR Imaging and Tractography Study—A Case Report and a Systematic Review"

_diagnostics, 2023, doi:10.3390/diagnostics13091602_

Round 1
Reviewer 1 Report
In Figure 3, using two different independent PDF readers, this review fails to find a legend representative of witch structures are indicated by the 4 different couples of bars reported in the caption. Furthermore the authors did not reported which bundle is reconstructed and superimposed on the CISS sequence (looks like the IFO but this reviewer fails to find a reference in the tex or in the caption about it) and the authors should also report which software was used for this particular reconstruction (looks they used MRI vendor workstation).
Line 74: "abdominal ultrasound" instead of "abdomen echography" should be considered togheter with other minor English language revision to the whole manuscript.
Line 79: the effective size of the spleen is not reported.
Lines 86-96: Please add acquisition times for each pulse sequence, the kind of coil used, and, since a SWAN sequence was scanned, it would be interesting to look at the intralesional vessel(s) on a minIP reconstruction instead of one of the images included in figure 2.
Line 103: FSL software version is missing (considering the different approach used in the latest versions of FSL for motion and EDDY current correction, this should be reported).
Line 126: The authors report "Only ATR and UF revealed a significant (p <0.001) difference between two side." Considering that the case report aims at a careful and rich iconography, I would suggest reviewing the figures, possibly adding three-dimensional (deterministic) reconstructions of the most involved fiber bundles together with the DTI altered metrics.
Minor English revision is suggested.
Author Response
Point 1. We correct fig. 3 adding a proper didascaly. We complete the text and the figure with the fiber bundle compressed.
Point 2. We revise some English language and expression.
Point 3. The effective spleen size was added.
Point 4. Acquisition time and other parameters was added for each sequence. The fig. 2 was improved with an extract of the SWAN acquisition.
Point 5. FSL software version was added.
Point 6. It was impossible to make an adequate statistical analysis considering the single case.
We are very grateful for your suggestions.
Reviewer 2 Report
Dear Authors,
The study presents an unusual case, rarely seen, of a very large Virchow-Robin space. Especially for imaging and neurosurgery professionals at the beginning of their careers, a case like this can be challenging (as it happened). That's why I appreciate your thorough imaging study besides the particularities of the case.
I would make one suggestion though. If possible I think the data in Tabel 2, which is hard to interpret as it is, could be plotted in a graph, giving a more visual and clear way of understanding the point it's trying to make.
Also, an emphasis on the differential diagnosis including imaging characteristics I believe would increase the appeal and the utility of the paper.
There are misspells and topic issues in some of the phrases that should be addressed.
Author Response
Point 1. We're sorry but it was very difficult for us to make a proper graphic explaining tab. 2.
Point 2. We add radiologic features in differential diagnosis, as you suggest. Thank you for your suggestions.